# DYNAMIC GRAPH: LEARNING INSTANCE-AWARE CONNECTIVITY FOR NEURAL NETWORKS

## ABSTRACT

One practice of employing deep neural networks is to apply the same architecture to all the input instances. However, a fixed architecture may not be representative enough for data with high diversity. To promote the model capacity, existing approaches usually employ larger convolutional kernels or deeper network structure, which may increase the computational cost. In this paper, we address this issue by raising the Dynamic Graph Network (DG-Net). The network learns the instance-aware connectivity, which creates different forward paths for different instances. Specifically, the network is initialized as a complete directed acyclic graph, where the nodes represent convolutional blocks and the edges represent the connection paths. We generate edge weights by a learnable module *router* and select the edges whose weights are larger than a threshold, to adjust the connectivity of the neural network structure. Instead of using the same path of the network, DG-Net aggregates features dynamically in each node, which allows the network to have more representation ability. To facilitate the training, we represent the network connectivity of each sample in an adjacency matrix. The matrix is updated to aggregate features in the forward pass, cached in the memory, and used for gradient computing in the backward pass. We verify the effectiveness of our method with several static architectures, including MobileNetV2, ResNet, ResNeXt, and RegNet. Extensive experiments are performed on ImageNet classification and COCO object detection, which shows the effectiveness and generalization ability of our approach.

## 1 INTRODUCTION

Deep neural networks have driven a shift from feature engineering to feature learning. The great progress largely comes from well-designed networks with increasing capacity of models (He et al., 2016a; Xie et al., 2017; Huang et al., 2017; Tan & Le, 2019). To achieve the superior performance, a useful practice is to add more layers (Szegedy et al., 2015) or expand the size of existing convolutions (kernel width, number of channels) (Huang et al., 2019; Tan & Le, 2019; Mahajan et al., 2018). Meantime, the computational cost significantly increases, hindering the deployment of these models in realistic scenarios. Instead of adding much more computational burden, we prefer adding *sample-dependent* modules to networks, increasing the model capacity by accommodating the data variance.

Several existing work attempt to augment the sample-dependent modules into network. For example, Squeeze-and-Excitation network (SENet) (Hu et al., 2018) learns to scale the activations in the *channel* dimension conditionally on the input. Conditionally Parameterized Convolution (CondConv) (Yang et al., 2019) uses over-parameterization weights and generates individual convolutional *kernels* for each sample. GaterNet (Chen et al., 2018) adopts a gate network to extract features and generate sparse binary masks for selecting *filters* in the backbone network based upon inputs. All these methods focus on the adjustment of the *micro* structure of neural networks, using a data-dependent module to influence the feature representation at the same level. Recall the deep neural network to mammalian brain mechanism in biology (Rauschecker, 1984), the neurons are linked by synapses and responsible for sensing different information, the synapses are activated to varying degrees when the neurons perceive external information. Such a phenomenon inspires us to design a data-dependent network structure so that different samples will activate different network paths.

In this paper, we learn to optimize the connectivity of neural networks based upon inputs. Instead of using stacked-style or hand-designed manners, we allow more flexible selection for forwarding paths.

Specifically, we reformulate the network into a directed acyclic graph, where nodes represent the convolution block while edges indicate connections. Different from randomly wired neural networks (Xie et al., 2019) that generate random graphs as connectivity using predefined generators, we rewire the graph as a complete graph so that all nodes establish connections with each other. Such a setting allows more possible connections and makes the task of finding the most suitable connectivity for each sample equivalent to finding the optimal sub-graph in the complete graph. In the graph, each node aggregates features from the preceding nodes, performs feature transformation (e.g. convolution, normalization, and non-linear operations), and distributes the transformed features to the succeeding nodes. The output of the last node in the topological order is employed as the representation through the graph. To adjust the contribution of different nodes to the feature representation, we further assign weights to the edges in the graph. The weights are generated dynamically for each input via an extra module (denoted as *router*) along with each node. During the inference, only crucial connections are maintained, which creates different paths for different instances. As the connectivity for each sample is generated through non-linear functions determined by routers, our method can enable the networks to have more representation power than the static network.

We call our method as the Dynamic Graph Network (DG-Net). It doesn't increase the depth or width of the network, while only introduces an extra negligible cost to compute the edge weights and aggregate the features. To facilitate the training, we represent the network connection of each sample as a adjacent matrix and design a buffer mechanism to cache the matrices of a sample batch during training. With the buffer mechanism, we can conveniently aggregate the feature maps in the forward pass and compute the gradient in the backward pass by looking up the adjacent matrices. The main contributions of our work are as follows:

- We *first* introduce the dynamic connectivity based upon inputs to exploit the model capacity of neural networks. Without bells and whistles, simply replacing static connectivity with dynamic one in many networks achieves solid improvement with only a slight increase of ($\sim 1\%$) parameters and ($\sim 2\%$) computational cost (see table 1).

- DG-Net is easy and memory-conserving to train. The parameters of networks and routers can be optimized in a differentiable manner. We also design a buffer mechanism to conveniently access the network connectivity, in order to aggregate the feature maps in the forward pass and compute the gradient in the backward pass.

- We show that DG-Net not only improves the performance for human-designed networks (e.g. MobielNetV2, ResNet, ResNeXt) but also boosts the performance for automatically searched architectures (e.g. RegNet). It demonstrates good generalization ability on ImageNet classification (see table 1) and COCO object detection (see table 2) tasks.

## 2 RELATED WORK

**Non-modular Network Wiring.** Different from the modularized designed network which consists of topologically identical blocks, there exists some work that explores more flexible wiring patterns. MaskConnect (Ahmed & Torresani, 2018) removes predefined architectures and learns the connections between modules in the network with $k$ conenctions. Randomly wired neural networks (Xie et al., 2019) use classical graph generators to yield random wiring instances and achieve competitive performance with manually designed networks. DNW (Wortsman et al., 2019) treats each channel as a node and searches a fine-grained sparse connectivity among layers. TopoNet (Yuan et al., 2020) learns to optimize the connectivity of neural networks in a complete graph that adapt to the specific task. Prior work demonstrates the potential of more flexible wirings, our work on DG-Net pushes the boundaries of this paradigm, by enabling each example to be processed with different connectivity.

**Dynamic Networks.** Dynamic networks, adjusting the network architecture to the corresponding input, have been recently studied in the computer vision domain. SkipNet (Wang et al., 2018b), BlockDrop (Wu et al., 2018) and HydraNet (Mullapudi et al., 2018) use reinforcement learning to learn the subset of blocks needed to process a given input. Some approaches prune channels (Lin et al., 2017a; You et al., 2019) for efficient inference. However, most prior methods are challenging to train, because they need to obtain discrete routing decisions from individual examples. Different from these approaches, DG-Net learns continuous weights for connectivity to enable ramous propagation of features, so can be easily optimized in a differentiable way.

**Conditional Attention.** Some recent work proposes to adapt the distribution of features or weights through attention conditionally on the input. SENet (Hu et al., 2018) boosts the representational power of a network by adaptively recalibrating channel-wise feature responses by assigning attention over channels. CondConv (Yang et al., 2019) and dynamic convolution (Chen et al., 2020) are restricted to modulating different experts/kernels, resulting in attention over convolutional weights. Attention-based models are also widely used in language modeling (Luong et al., 2015; Bahdanau et al., 2015; Vaswani et al., 2017), which scale previous sequential inputs based on learned attention weights. In the vision domain, previous methods most compute attention over *micro* structure, ignoring the influence of the features produced by different layers on the final representation. Unlike these approaches, DG-Net focuses on learning the connectivity based upon inputs, which can be seen as attention over features with different semantic hierarchy.

**Neural Architecture Search.** Recently, Neural Architecture Search (NAS) has been widely used for automatic network architecture design. With evolutionary algorithm (Real et al., 2019), reinforcement learning (Pham et al., 2018) or gradient descent (Liu et al., 2019), one can obtain task-dependent architectures. Different from these NAS-based approaches, which search for a single architecture, the proposed DG-Net generates forward paths on the fly according to the input without searching. We also notice a recent method InstaNAS (Cheng et al., 2020) that generates domain-specific architectures for different samples. It trained a controller to select child architecture from the defined meta-graph, achieving latency reduction during inference. Different from them, DG-Net focuses on learning connectivity in a complete graph using a differentiable way and achieves higher performance.

## 3 METHODOLOGY

### 3.1 NETWORK REPRESENTATION WITH DAGs

The architecture of a neural network can be naturally represented by a *directed acyclic graphs (DAG)*, consisting of an ordered sequence of nodes. Specifically, we map both combinations (e.g., addition) and transformation (e.g., convolution, normalization, and activation) into a node. At the same time, connections between layers are represented as edges, which determine the path of the features in the network. For simplicity, we denote a DAG with $N$ ordered nodes as $\mathcal{G} = (\mathcal{N}, \mathcal{E})$, where $\mathcal{N}$ is the set of nodes and $\mathcal{E}$ is the set of edges. We show $\mathcal{E} = \{e^{(i,j)} | 1 \leq i < j \leq N\}$, where $e^{(i,j)}$ indicates a directed edge from the $i$-th node to the $j$-th node.

Most traditional convolutional neural networks can be represented with DAGs. For example, VGGNet (Simonyan & Zisserman, 2015) is stacked directly by a series of convolutional layers, where a current layer is only connected to the previous layer. The connectivity in each stage can be represented as $\mathcal{E}_{vgg} = \{e^{(i,j)} | j = i + 1 |_{1 \leq i < N}\}$. To ease problems of gradient vanishing and exploding, ResNets (He et al., 2016a) build additional shortcut and enable cross-layer connections whose nature view [1] can be denoted by $\mathcal{E}_{res} = \{e^{(i,j)} | j \in \{i + 1, i + 2\} |_{1 \leq i < N}\}$. It is worth noting that some NAS methods (Real et al., 2019; Liu et al., 2019) also follow this wiring pattern that blocks connect two immediate preceding blocks. Differently, DenseNets (Huang et al., 2017) aggregate features from all previous layers in the manner of $\mathcal{E}_{dense} = \{e^{(i,j)} | i \in [1, j - 1] |_{1 < j \leq N}\}$. Given these patterns of connectivity, the forward procedure of network can be performed according to the topological order. For the $j$-th node, the output feature $\mathbf{x}^{(j)}$ is computed by:

$$\mathbf{x}^{(j)} = f^{(j)}(\sum_{i<j} \mathbb{1}_{\mathcal{E}}(e^{(i,j)}) \cdot \mathbf{x}^{(i)}), \ s.t. \ \mathbb{1}_{\mathcal{E}}(e^{(i,j)}) \in \{0, 1\} \tag{1}$$

where $f^{(j)}(\cdot)$ is the corresponding mapping function for transformations, and $\mathbb{1}_{\mathcal{E}}(e^{(i,j)})$ stands for the indicator function and equals to one when $e^{(i,j)}$ exists in $\mathcal{E}$.

In each graph, the first node in topological order is the input one that only performs the distribution of features. The last node is the output one that only generates final output by gathering preceding inputs. For a network with $K$ stages, $K$ DAGs are initialized and connected sequentially. Each graph is linked to its preceding or succeeding stage by output or input node. Let $\mathcal{F}^{(k)}(\cdot)$ be the mapping

---

[1]In (Veit et al., 2016), its unrolled type can be viewed as $\mathcal{E}_{dense} = \{e^{(i,j)} | i \in [1, j - 1] |_{1 < j \leq N}\}$.

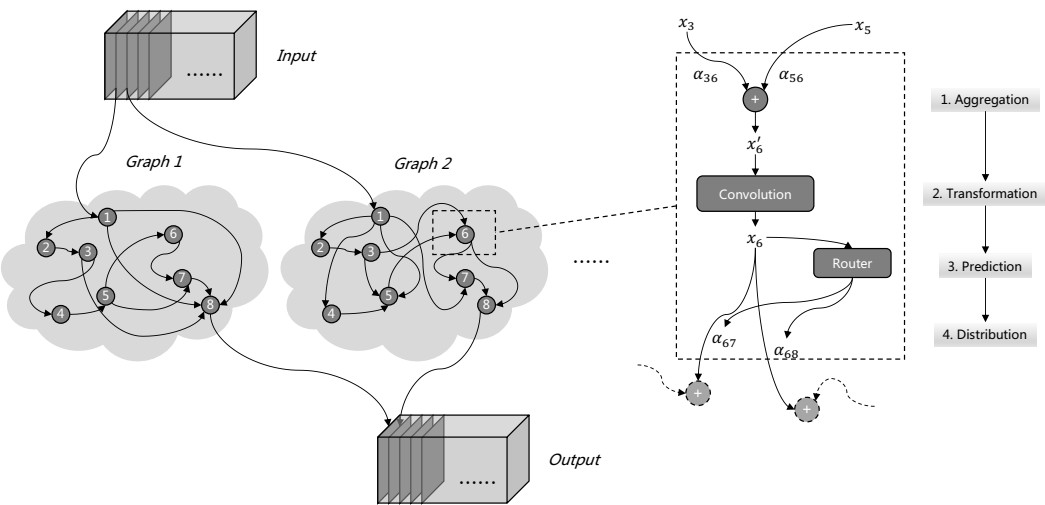

Figure 1: The framework of DG-Net. *Left*: For a training batch, each sample performs different forward paths that are determined by the sample-dependent macro connectivity. *Right*: Node operations at the micro level. Here we illustrate a node with 2 active input edges and output edges. First, it aggregates input features from preceding nodes by weighted sum. Second, convolutional blocks transform the aggregated features. Third, a router predicts routing weights for each sample on the output edges according to the transformed features. Last, the transformed data is sent out by the output edges to the following nodes. Arrows indicate the data flow.

function of the $k$-th stage, which is established by $\mathcal{G}^{(k)}$ with nodes $\mathcal{N}^{(k)}$ and connectivity $\mathcal{E}^{(k)}$. Given an input $\mathbf{x}$, the mapping function from the sample to the feature representation can be written as:

$$\mathcal{T}(\mathbf{x}) = \mathcal{F}^{(K)}(\cdots \mathcal{F}^{(2)}(\mathcal{F}^{(1)}(\mathbf{x})))$$ (2)

### 3.2 EXPANDING SEARCH SPACE FOR THE CONNECTIVITY

As shown in Eq.(1), most traditional networks adopt binary codes to formulate the connectivity, resulting in a relatively sparse and static connection pattern. But these prior-based methods limit the connection possibilities, the type of feature fusion required by different samples may be different. In this paper, we raise two modifications in DG-Net to expand the search space with more possible connectivity. *First*, we remove the constraint on the in/out-degree of nodes and initialize the connectivity to be a *complete* graph where edges exist between any nodes. The search space is different from DenseNet that we replace the aggregation method from concatenation to addition. This avoids misalignment of channels caused by the removal or addition of edges. In this way, finding good connectivity is akin to finding optimal sub-graphs. *Second*, instead of selecting discrete edges in the binary type, we assign a soft weight $\alpha^{(i,j)}$ to the edge which reflects the magnitude of connections. This also benefits the connectivity can be optimized in a differentiable manner.

In neural networks, features generated by different layers exhibit various semantic representations (Zhou et al., 2016; Zeiler & Fergus, 2014). Recall to the mammalian brain mechanism in biology (Rauschecker, 1984) that the synapses are activated to varying degrees when the neurons perceive external information, the weights of edges in the graph can be parameterized upon inputs. As shown in the left of Fig. 1, DG-Net can generate appropriate connectivity for each sample. Different from Eq.(1), the output feature can be computed by:

$$\mathbf{x}^{(j)} = f^{(j)}\Big(\sum_{i<j} \boldsymbol{\alpha}^{(i,j)} \cdot \mathbf{x}^{(i)}\Big)$$ (3)

where $\boldsymbol{\alpha}^{(i,j)}$ is a vector that contains the weights related to samples in a batch.

### 3.3 INSTANCE-AWARE CONNECTIVITY THROUGH ROUTING MECHANISM

To obtain $\alpha^{(i,j)}$ and allow instance-aware connectivity for the network, we add an extra conditional *router* module along with each node, as presented in the right of Fig. 1. The calculation procedure in a node can be divided into four steps. *First*, the node aggregates features from preceding connected nodes by weighted addition. *Second*, the node performs feature transformation with convolution, normalization, and activation layers (determined by the network). *Third*, the router receives the transformed feature and applies squeeze-and-excitation to compute instance-aware weights over edges with succeeding nodes. *Last*, the node distributes the transformed features to succeeding nodes according to the weights. We also discuss the different implementation of routers in appendix 6.2. This method is equivalent to building an independent router for each edge.

Structurally, the router applies a lightweight module consisting of a *global average pooling* $\phi(\cdot)$, a *fully-connected layer* and a *sigmoid activation* $\sigma(\cdot)$. The global spatial information is firstly squeezed by global average pooling. Then we use a fully-connected layer and sigmoid to generate normalized routing weights $\alpha^{(i,j)}$ for output edges. The mapping function of the router can be written as:

$$\varphi(\mathbf{x}) = \sigma(\mathbf{w}^T \phi(\mathbf{x}) + \mathbf{b}), \ s.t. \ \varphi(\cdot) \in [0, 1), \tag{4}$$

where $\mathbf{w}$ and $\mathbf{b}$ are weights and bias of the fully-connected layer. Particularly, DG-Net is computationally efficient because of the 1-D dimension reduction of $\phi(\cdot)$. For an input feature map with dimension $H \times W \times C_{in}$, the convolutional operation requires $C_{in}C_{out}HWD_k^2$ Multi-Adds (for simplicity, only one layer of convolution is calculated), where $D_k$ is the kernel size. As for the routing mechanism, it only introduces extra $O(\varphi(\mathbf{x})) = C_{in}\zeta_{out}$ Multi-Adds, where $\zeta_{out}$ is the number of output edges for the node. This is much less than the computational cost of convolution.

Besides, we set a learnable weight of $\tau$ that acts as a threshold for each node to control the connectivity. When the weight is less than the threshold, the connection will be closed during inference. When $\alpha^{(i,j)} = 0$, the edge from $i$-th node to $j$-th node will be marked as closed. If the input or output edges for a node are all closed, the node will be removed to accelerate inference time. Meanwhile, all the edges with $\alpha^{(i,j)} > 0$ will be reserved, continuously enabling feature fusion. This can be noted by:

$$\alpha^{(i,j)} = \left\{ \begin{array}{ll} 0 & \alpha^{(i,j)} < \tau \\ \alpha^{(i,j)} & \alpha^{(i,j)} \geq \tau \end{array} \right. . \tag{5}$$

During training, this can be implemented by a differentiable manner of $\psi(\alpha) = \alpha \cdot \sigma(\alpha - \tau)$.

### 3.4 BUFFER MECHANISM FOR MACRO FEATURE AGGREGATION

DG-Net allows flexible wiring patterns for the connectivity, which requires the aggregation of the features among nodes that need to be recorded and shared within a graph. For this purpose, we represent the connectivity in an adjacency matrix (denoted as $\mathbf{M} \in \mathbb{R}^{N \times N}$). The order of rows and columns indicate the topological order of the nodes in the graph. Elements in the matrix represent the weights of edges, as shown in the left of Fig. 2, where rows reflect the weights of input edges and columns are of output edges for a node. During the forward procedure, the $i$-th node performs aggregation through weights *acquired* from the corresponding row of $\mathbf{M}_{i-1,:}$. Then the node generates weights over output edges through accompanying the router and *stores* them into the column of $\mathbf{M}_{:,i}$. In this way, the adjacency matrix is updated progressively and shared within the graph. For a batch with $B$ samples, different matrices are concatenated in the dimension of batch and cached in a defined buffer (denoted as $\mathbf{M} \in \mathbb{R}^{B \times N \times N}$, where $\mathbf{M}_{b,:,:} = \mathbf{M}$), as shown in the right of Fig. 2. With the buffer mechanism, DG-Net can be trained as an ordinary network without introducing excessive computation or time-consuming burden.

### 3.5 OPTIMIZATION OF DG-NET

During training, the parameters of the network $\mathbf{W}_n$, as well as the parameters of routers $\mathbf{W}_r$, are optimized simultaneously using gradients back-propagation. Given an input $\mathbf{x}$ and corresponding label $\mathbf{y}$, the objective function can be represented as:

$$\min_{\mathbf{W}_n, \mathbf{W}_r} \mathcal{L}_t(\mathcal{T}(\mathbf{x}; \mathbf{W}_n, \mathbf{W}_r), \mathbf{y}) \tag{6}$$

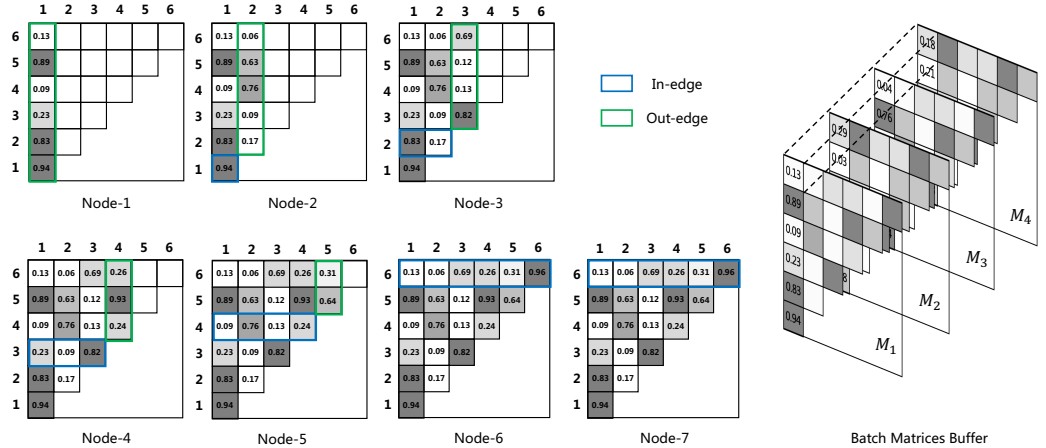

Figure 2: The procedure of updating the adjacency matrix and the proposed buffer for storing. A node obtains the weights of input edges from the row (blue) and storing weights to output edges saving in the column (green). The matrices are saved in a buffer that supports batch training efficiently.

where $\mathcal{L}_t(\cdot, \cdot)$ denotes the loss function w.r.t specific tasks (e.g. cross-entropy loss for image classification and regression loss for object detection). This method has two benefits. First, simultaneous optimization can effectively reduce the time consumption of training. The time to obtain a trained dynamic network is the same as that of a static network. Second, different from DARTS (Liu et al., 2019) that selects operation with the maximum probability, our method learns the connectivity in a continuous manner, which better preserves the consistency between training and testing.

Set $\frac{\partial \mathcal{L}_t}{\partial \mathbf{w}_n^j}$ be the gradients that the network flows backwards to the convolutional weights of the $j$-th node $\mathbf{w}_n^{(j)}$. Let $\frac{\partial \mathcal{L}_t}{\partial \mathbf{x}^j}$ be the gradients to $\mathbf{x}^{(j)}$. Then the gradients update to the weights of node $\mathbf{w}_n^{(j)}$, the weights of router $\mathbf{w}_r^{(j)}$, the biases of router $\mathbf{b}_r^{(j)}$ and threshold $\tau^{(j)}$ are of the form:

$$\mathbf{w}_n^{(j)} \quad \leftarrow \quad \mathbf{w}_n^{(j)} - \eta \cdot \frac{\partial \mathcal{L}_t}{\partial \mathbf{w}_n^j} \tag{7}$$

$$\mathbf{w}_r^{(i,j)} \quad \leftarrow \quad \mathbf{w}_r^{(i,j)} - \eta \cdot \sum \left( \frac{\partial \mathcal{L}_t}{\partial \mathbf{x}^j} \odot \frac{\partial f^j}{\partial \mathbf{x}^{j'}} \odot \mathbf{x}^i \right) \cdot \frac{\partial \varphi^j}{\partial z^i} \cdot \phi^j(\mathbf{x}^i) \tag{8}$$

$$b_r^{(i,j)} \quad \leftarrow \quad b_r^{(i,j)} - \eta \cdot \sum \left( \frac{\partial \mathcal{L}_t}{\partial \mathbf{x}^j} \odot \frac{\partial f^j}{\partial \mathbf{x}^{j'}} \odot \mathbf{x}^i \right) \cdot \frac{\partial \varphi^j}{\partial z^i} \tag{9}$$

$$\tau^{(j)} \quad \leftarrow \quad \tau^{(j)} - \eta \cdot \sum \left( \frac{\partial \mathcal{L}_t}{\partial \mathbf{x}^j} \odot \frac{\partial f^j}{\partial \mathbf{x}^{j'}} \odot \mathbf{x}^i \right) \cdot \frac{\partial \psi^j}{\partial \tau^j} \tag{10}$$

where $\mathbf{w}_r^{(i,j)} \in \mathbb{R}^{C \times 1}$ and $b_r^{(i,j)} \in \mathbb{R}^1$ are the weights and bias of the router that determine the output edge of $e^{(i,j)}$. And $\mathbf{x}^{j'}$ is the aggregated features of $\sum \alpha^{(i,j)} \cdot \mathbf{x}^{(i)}$ in Eqn. (3), $z^i$ is calculated by ${\mathbf{w}^{(i,j)}}^T \phi(\mathbf{x}^i) + b^{(i,j)}$ in Eqn. (4). And $\frac{\partial \psi^j}{\partial \tau^j}$ is $\alpha^{ij} \cdot \sigma(\alpha^{ij} - \tau^j) \cdot (\sigma(\alpha^{ij} - \tau^j) - 1)$. And $\odot$ indicates entrywise product. The gradients w.r.t $\alpha^{(i,j)}$ can be noted as $\sum \left( \frac{\partial \mathcal{L}_t}{\partial \mathbf{x}^j} \odot \frac{\partial f^j}{\partial \mathbf{x}^{j'}} \odot \mathbf{x}^i \right)$.

# 4 EXPERIMENTS

## 4.1 IMAGENET CLASSIFICATION

**Dataset and evaluation metrics.** We evaluate our approach on the ImageNet 2012 classification dataset (Russakovsky et al., 2015). The ImageNet dataset consists of 1.28 million training images and 50,000 validation images from 1000 classes. We train all models on the entire training set and compare the single-crop top-1 validation set accuracy with input image resolution 224×224. We measure performance as ImageNet top-1 accuracy relative to the number of parameters and computational cost in FLOPs.

**Model selection and training setups.** We validate our approach on a number of widely used models including MobileNetV2-1.0 (Sandler et al., 2018), ResNet-18/50/101 (He et al., 2016a) and ResNeXt50-32x4d (Xie et al., 2017). To further test the effectiveness of DG-Net, we attempt to optimize recent NAS-based networks of RegNets (Radosavovic et al., 2020), which are the best models out of a search space with $\sim 10^{18}$ possible configurations. Our implementation is based on PyTorch (Paszke et al., 2019) and all experiments are conducted using 16 NVIDIA Tesla V100 GPUs with a total batch of 1024. All models are trained using SGD optimizer with 0.9 momentum. Detailed information about the training setting can be seen in appendix 6.1.

Table 1: ImageNet validation accuracy (%) and inference cost. DG-Nets improve the accuracy of all baseline architectures with small relative increase in the number of parameters and inference cost.

| Network | Baselines | | | DG-Net | | | $\Delta$ Top-1 |
|---|---|---|---|---|---|---|---|
| | Params(M) | FLOPs(M) | Top-1 | Params(M) | FLOPs(M) | Top-1 | |
| MobileNetV2-1.0 | 3.51 | 299 | 72.60 | 3.58 | 312 | **73.54** | + 0.94 |
| ResNet18 | 11.69 | 1813 | 70.30 | 11.71 | 1826 | **71.32** | + 1.02 |
| ResNet50 | 25.55 | 4087 | 76.70 | 25.62 | 4125 | **78.28** | + 1.58 |
| ResNet101 | 44.54 | 7799 | 78.29 | 44.90 | 7837 | **79.90** | + 1.61 |
| ResNeXt50-32x4d | 25.02 | 4228 | 77.97 | 25.09 | 4305 | **79.39** | + 1.42 |
| RegNet-X-600M | 6.19 | 599 | 74.03 | 6.22 | 600 | **74.68** | + 0.65 |
| RegNet-X-1600M | 9.19 | 1602 | 77.26 | 9.22 | 1604 | **77.91** | + 0.65 |

**Analysis of experimental results.** We verify that DG-Net improves performance on a wide range of architectures in Table 1. For fair comparison, we retrain all of our baseline models with the same hyperparameters as the DG-Net models[2]. Compared with baselines, DG-Net gets considerable gains with a small relative increase in the number of parameters ($< 2\%$) and inference cost of FLOPs ($< 1\%$). This includes architectures with mobile setting (Sandler et al., 2018), classical residual wirings (He et al., 2016a; Xie et al., 2017), multi-branch operation (Xie et al., 2017) and architecture search (Radosavovic et al., 2020). We further find that DG-Net benefits from the large search space which can be seen in the improvements of ResNets. With the increase of the depth from 18 to 101, the formed complete graph includes more nodes, resulting in larger search space and more possible connectivity. And the gains raise from $1.02\%$ to $1.61\%$ in top-1 accuracy. Ablation study on different connectivity method are given in appendix 6.3.

## 4.2 COCO OBJECT DETECTION

We report the transferability results by fine-tuning the networks for COCO object detection (Lin et al., 2014). We use Faster R-CNN (Ren et al., 2015) with FPN (Lin et al., 2017b) as the object detector. Our fine-tuning is based on the $1\times$ setting of the publicly available `Detectron2` (Girshick et al., 2018). We replace the backbone with those in Table 1.

The object detection results are given in Table 2. And FLOPs of the backbone are computed with an input size of 800×1333. Compared with the static network, DG-Net improves AP by $1.70\%$ with ResNet-50 backbone. When using a larger search space of ResNet101, our method significantly improves the performance by $2.73\%$ in AP. It is worth noting that stable gains are obtained for objects of different scales varying from small to large. This further verifies that instance-aware connectivity can improve the representation capacity toward the dataset with a large distribution variance.

---

[2]Our re-implementation of the baseline models and our DG-Net models use the same hyperparameters. For reference, published results for baselines are: MobileNetV2-1.0 (Sandler et al., 2018): 72.00%, ResNet-18 (He et al., 2016b), 69.57%, ResNet-50 (Goyal et al., 2017): 76.40%, ResNet-101 (Goyal et al., 2017): 77.92%, ResNeXt50-32x4d (Xie et al., 2017): 77.80%, RegNetX-600M (Radosavovic et al., 2020): 74.10%, RegNetX-1600M (Radosavovic et al., 2020): 77.00%.

Table 2: COCO object detection *minival* performance. APs (%) of bounding box detection are reported. DG-Nets brings consistently improvement across multiple backbones on all scales.

| Backbone | Method | GFLOPs | AP | $AP_{.5}$ | $AP_{.75}$ | $AP_S$ | $AP_M$ | $AP_L$ |
|---|---|---|---|---|---|---|---|---|
| ResNet50 | Baseline | 174 | 36.42 | 58.54 | 39.11 | 21.93 | 40.02 | 46.58 |
| | DG-Net | 176 | $38.12_{+1.70}$ | 60.53 | 41.00 | 23.61 | 41.52 | 48.39 |
| ResNet101 | Baseline | 333 | 38.59 | 60.56 | 41.63 | 22.45 | 43.08 | 49.46 |
| | DG-Net | 335 | $41.32_{+2.73}$ | 63.54 | 44.97 | 25.71 | 45.60 | 52.62 |
| ResNeXt50-32x4d | Baseline | 181 | 38.07 | 60.42 | 41.01 | 22.97 | 42.10 | 48.68 |
| | DG-Net | 183 | $39.52_{+1.45}$ | 62.41 | 42.56 | 25.71 | 43.34 | 49.83 |

## 4.3 COMPARED WITH RELATED WORK

**Compared with InstaNAS (Cheng et al., 2020).**  InstaNAS generates data-dependent networks from a designed meta-graph. During inference, it uses a controller to sample possible architectures by a Bernoulli distribution. But it needs to carefully design the training process to avoid collapsing the controller. Differently, DG-Net builds continuous connections between nodes, which allowing more possible connectivity. And the proposed method is compatible with gradient descent, and can be trained in a differentiable way easily. MobileNetV2 is used as the backbone network in InstaNAS. It provides multiple searched architectures under different latencies. For a fair comparison, DG-Net adopts the same structure as the backbone and reports the results of ImageNet. The latency is tested using the same hardware. The results in Table 3 demonstrate DG-Net can generate better instance-aware architectures in the dimension of connectivity.

**Compared with RandWire (Xie et al., 2019).**  Randomly wired neural networks explore using flexible graphs generated by different graph generators as networks, losing the constraint on wiring patterns. But for the entire dataset, the network architecture it uses is still consistent. Furthermore, DG-Net allows instance-aware connectivity patterns learned from the complete graph. We compare three types of generators in their paper with best hyperparameters, including Erdös-Rényi (ER), Barabási-Albert (BA), and Watts-Strogatz (WS). Since the original paper does not release codes, we reproduce these graphs using `NetworkX`[3]. We follow the small computation regime to form networks. Experiments are performed in ImageNet using its original training setting except for the DropPath and DropOut. Comparison results are shown in Table 4. DG-Net is superior to three classical graph generators in a similar computational cost. This proves that under the same search space, the optimized data-dependent connectivity is better than randomly wired static connectivity.

Table 3: Compared with InstaNAS under comparable latency in ImageNet.

| Model | Top-1 | Latency (ms) |
|---|---|---|
| InstaNAS-ImgNet-A | 71.9 | $0.239_{\pm0.014}$ |
| InstaNAS-ImgNet-B | 71.1 | $0.189_{\pm0.012}$ |
| InstaNAS-ImgNet-C | 69.9 | $0.171_{\pm0.011}$ |
| DG-Net-MBv2-1.0 | $\mathbf{73.5}_{\pm0.06}$ | $0.257_{\pm0.015}$ |

Table 4: Compared with RandWire under small computation regime in ImageNet.

| Wiring Type | Top-1 | FLOPs(M) |
|---|---|---|
| ER (P=0.2) | $71.34_{\pm0.40}$ | 602 |
| BA (M=5) | $71.16_{\pm0.34}$ | 582 |
| WS (K=4, P=0.75) | $72.26_{\pm0.27}$ | 572 |
| DG-Net | $\mathbf{73.52}_{\pm0.05}$ | 611 |

**Compared with state-of-the-art NAS-based methods.**  For completeness, we compare with the most accurate NAS-based networks under the mobile setting ($\sim$ 600M FLOPs) in ImageNet. It is worth noting that this is *not* the focus of this paper. We select RegNet as the basic architecture as shown in Table 1. For fair comparisons, here we train 250 epochs and other settings are the same with section 4.1. We note RegNet-X with the dynamic graph as DG-Net-A and RegNet-Y with a dynamic graph as DG-Net-B [4] (with SE-module for comparison with particular searched architectures e.g. EfficientNet). The experimental results are given in Table 5. It shows that with a single operation

---

[3] https://networkx.github.io

[4] The original performance of RegNet-X-600M is 75.03%, and RegNet-Y-600M is 76.10% under this training setting.

type (*Regular Bottleneck*), DG-Net can obtain considerable performance with other NAS methods with less search cost.

Table 5: Comparision with NAS methods under mobile setting. Here we train for 250 epochs similar to (Zoph et al., 2018; Real et al., 2019; Xie et al., 2019; Liu et al., 2018; 2019), for fair comparisons.

| Network | Params(M) | FLOPs(M) | Search Cost | Top-1 | Top-5 |
|---|---|---|---|---|---|
| NASNet-A (Zoph et al., 2018) | 5.3 | 564 | 2000 | 74.0 | 91.6 |
| NASNet-B (Zoph et al., 2018) | 5.3 | 488 | 2000 | 72.8 | 91.3 |
| NASNet-C (Zoph et al., 2018) | 4.9 | 558 | 2000 | 72.5 | 91.0 |
| Amoeba-A (Real et al., 2019) | 5.1 | 555 | 3150 | 74.5 | 92.0 |
| Amoeba-B (Real et al., 2019) | 5.3 | 555 | 3150 | 74.0 | 91.5 |
| RandWire-WS (Xie et al., 2019) | 5.6 | 583 | - | 74.7 | 92.2 |
| PNAS (Liu et al., 2018) | 5.1 | 588 | $\sim$225 | 74.2 | 91.9 |
| DARTS (Liu et al., 2019) | 4.9 | 595 | 4 | 73.1 | 91.0 |
| EfficientNet-B0 (Tan & Le, 2019) | 5.3 | 390 | - | 76.3 | 93.2 |
| DG-Net-A | 6.2 | 601 | 1.5 | 75.8 | 92.6 |
| DG-Net-B | 6.3 | 601 | 1.5 | 77.0 | 93.4 |

## 5 CONCLUSION AND FUTURE WORK

In this paper, we present the Dynamic Graph Network (noted as DG-Net), that allows learning instance-aware connectivity for neural networks. Without introducing much computation cost, the model capacity can be increased to ease the difficulties of feature representation for data with high diversity. We show that DG-Net is superior to many static networks, including human-designed and automatically searched architectures. Besides, DG-Net demonstrates good generalization ability on ImageNet classification as well as COCO object detection. It also achieved SOTA results compared with related work. DG-Net explores the connectivity in an enlarged search space, which we believe is a new research direction. In future work, we consider verifying DG-Net on more NAS-searched architectures. Moreover, we will study learning dynamic operations beyond the connectivity as well as adjusting the computation cost based upon the difficulties of samples.

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

## 6   APPENDIX

### 6.1   MODEL SETTINGS IN IMAGENET CLASSIFICATION

**Setup for MobileNetV2.**   We train the networks for 200 epochs with a half-period-cosine-shaped learning rate decay. The initial learning rate is 0.4 with a warmup phase of 5 epochs. The weight decay is set to 4e-5. To prevent overfitting, we use label smoothing (Szegedy et al., 2017) with a coefficient of 0.1 and dropout (Srivastava et al., 2014) before the last layer with rate of 0.2. For building DG-Net, the *Inverted Bottleneck* block is represented as a node.

**Setup for ResNets and ResNeXt.**   The networks are trained for 100 epochs with a half-period-cosine-shaped learning rate decay. The initial learning rate is 0.4 with a warmup phase of 5 epochs. The weight decay is set to e-4. We use label smoothing with a coefficient of 0.1. Other details of the training procedure are the same as (Goyal et al., 2017). To form DG-Net, the *BasicBlock* of ResNet18 and *Bottleneck* block of ResNet50/101 is denoted as nodes. For ResNeXt, the *Aggregated Bottleneck* block is set to be a node.

**Setup for RegNets.**   The networks are trained for 100 epochs with a half-period-cosine-shaped learning rate decay. The initial learning rate is 0.8 with a warmup phase of 5 epochs. The weight decay is set to 5e-5. We use label smoothing with a coefficient of 0.1. Other details are the same as (Radosavovic et al., 2020). For DG-Net, the *Regular Bottleneck* is transformed to be a node.

### 6.2   LOCATION OF ROUTERS

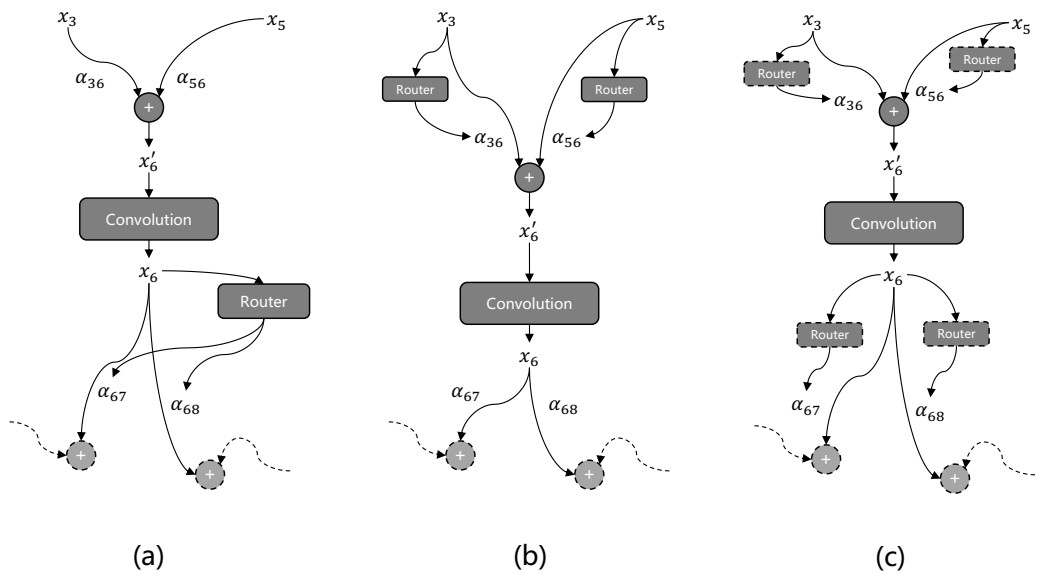

Figure 3: Different routing methods with different locations of routers.

In this paper, the connectivity is determined by the weights of edges which are predicted using extra conditional module routers. Within a node, there are two ways to obtain the weights, respectively predicting the weights of the output edges (as shown in Fig. 3 (a)) or predicting the weights of the input edges (as shown in Fig. 3 (b)). For the output type, as described in section 3.3, the router receives the transformed features as input and generates attention over output edges connected with posterior nodes. For the input type, the number of routers is related to the in-degree of the current node. Each router receives features from the connected preceding node and predicts weight for each input edge independently.

Although the form seems to be different, the two methods are equivalent under the routing function we designed. The router module consists of a global average pooling, a fully-connected layer and a

sigmoid activation. The router in Fig. 3 (a) can be split into the type of Fig. 3 (c). It can be noted as:

$$\varphi(\mathbf{g}) = \sigma(\mathbf{w}^T\mathbf{g} + \mathbf{b}) = [\sigma(\mathbf{w}_{:,0}^T\mathbf{g} + b_0), \cdots, \sigma(\mathbf{w}_{:,j}^T\mathbf{g} + b_j)] \tag{11}$$

where $\mathbf{g}$ is the feature vector after global average pooling, $\mathbf{w}_{:,j}$ is the weight of independent fully-connected layer after splitting, and $[\cdot]$ indicates concatenation. The prediction for the output edge of the current node is equal to the prediction for the input edge of the next node. Therefore, the two methods are equivalent. To simplify, we select the first type in implementation.

### 6.3 ABLATION STUDY ON CONNECTIVITY METHOD

We conduct an ablation study on different connectivity methods to reflect the effectiveness of the proposed DG-Net. The experiments are performed in ImageNet and follow the training setting in section 4.1. For a fair comparison, we select ResNet-50/101 as the backbone structure. The symbol $\alpha$ denotes assigning learnable parameters to the edge directly, which learns fixed connectivity for all samples. The symbol $\alpha_b$ denotes the type of DG-Net, which learns instance-aware connectivity. The experimental results are given in Table 6. In this way, ResNet-50 with $\alpha_b$ still outperforms one with $\alpha$ by 1.28% in top-1 accuracy. And ResNet-101 is the same. This demonstrates that due to the enlarged optimization space, dynamic connectivity is better than static connectivity in these networks.

Table 6: Ablation study on different connectivity methods. Experiments are conducted in ImageNet and Top-1 accuracy are reported. Results show that the proposed dynamic graph outperforms static networks with/without learnable weights of edges in large marigins.

| Backbone | $\alpha$ | $\alpha_b$ | Top-1 | $\Delta$ Top-1 |
|---|---|---|---|---|
| ResNet-18 | | | 70.30 | - |
| | ✓ | | 70.51 | + 0.21 |
| | | ✓ | 71.32 | + 1.02 |
| ResNet-50 | | | 76.70 | - |
| | ✓ | | 77.00 | + 0.30 |
| | | ✓ | 78.28 | + 1.58 |
| ResNet-101 | | | 78.29 | - |
| | ✓ | | 78.64 | + 0.35 |
| | | ✓ | 79.90 | + 1.61 |
| MobileNetV2-1.0 | | | 72.60 | - |
| | ✓ | | 72.86 | + 0.26 |
| | | ✓ | 73.54 | + 0.94 |

### 6.4 ABLATION STUDY ON INITIALIZATION SCHEMES FOR ROUTERS

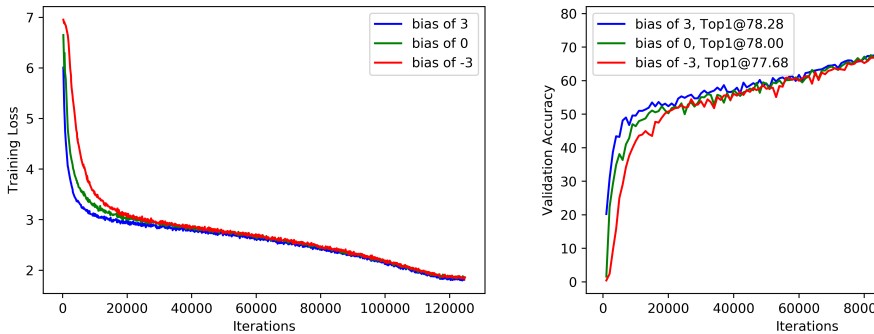

Figure 4: Different initialization schemes for routers. The positive bias initializes the connections as existence, obtains lower training loss in the early training procedure, and achieves higher validation accuracy than negative and zero biases.

For the routers, we use the routing transformation defined as $\varphi(\mathbf{x}) = \sigma(\mathbf{w}^T\phi(\mathbf{x}) + \mathbf{b})$, where $\mathbf{w}^T$ is the weight matrix and $\mathbf{b}$ the bias vector. This suggests a simple initialization scheme that the bias

can be initialized with a positive value (e.g. 3 etc.) such that the network is initially biased towards *existence connections* behavior. This scheme is strongly inspired by the proposal of (Gers et al., 2000) to initially bias the gates in a Long Short-Term Memory recurrent network to help bridge long-term temporal dependencies early in learning. And this initialization scheme is also adpoted in Highway netowrks (Srivastava et al., 2015) and Non-local networks (Wang et al., 2018a).

We conduct ablation study using DG-Net based on ResNet-50 in ImageNet. Details of the training procedure are the same in section 4.1. The bias is initialize with $\{3, 0, -3\}$ respectively. These initialization methods correspond to *existence connections*, *unbiased* and *non-existent connections*. Experimental results are given in Fig. 4. It can be seen that the positive initialization of bias achieves lower training loss in the early training procedure and obtains higher validation Top-1 accuracy of 78.28%. This suggests that initializing the connections to existing is better than unbiased initialization and non-existent.

## 6.5 STATISTICAL AND VISUALIZATION ANALYSES OF THE LEARNED CONNECTIVITY

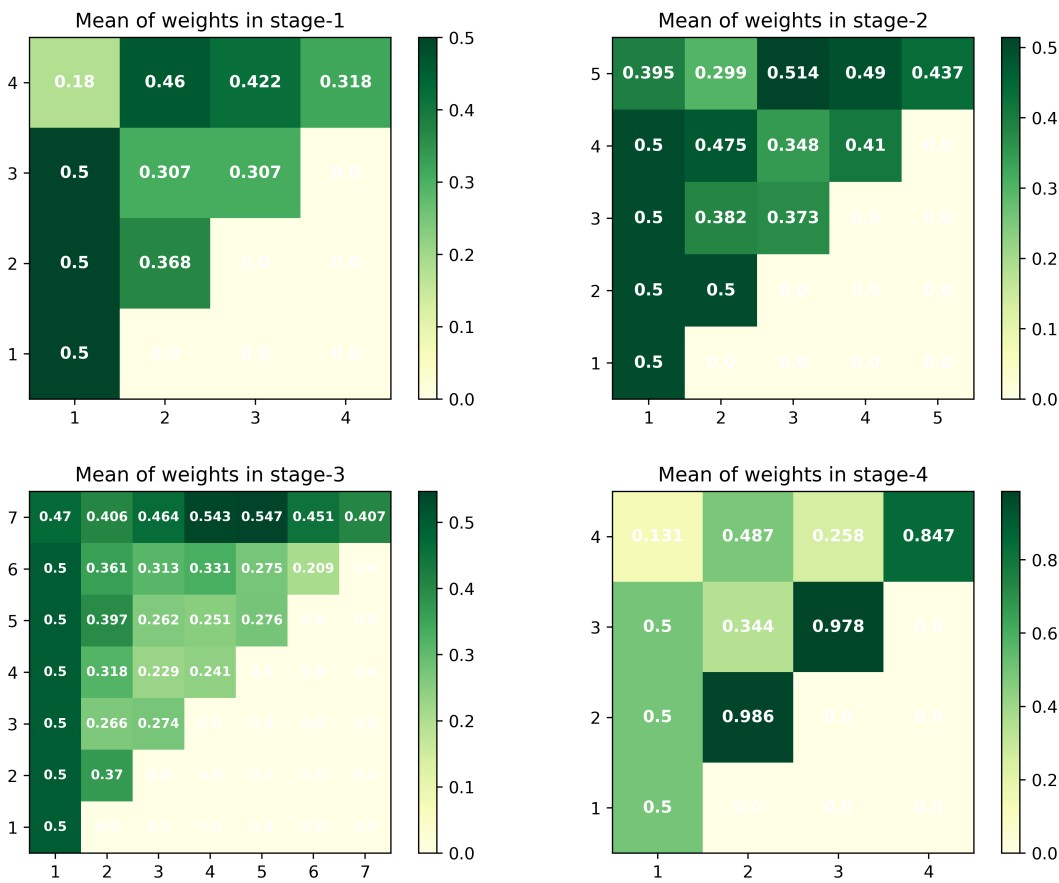

Figure 5: The distribution of mean of weights of edges in different graphs/stages. Darker colors represent larger weights.

To analyze the architecture representations, we visualize the learned connectivity through the adjacency matrices as noted in section 3.4. The validation dataset that contains 50000 images of ImageNet is used for inference. We select the trained DG-Net with ResNet-50 that contains 4 stages for analysis.

For statistical analysis, we show the distribution of the mean of the weights of edges in Fig. 5 and the distribution of standard deviation in Fig. 6. To visually analyze the connections corresponding to

different samples, we give the learned connectivity in Fig. 7. Some observations and analysis can be made:

(1) From the statistical analysis in Fig. 5, the weights of connections between different nodes have obvious differences. The difference is related to the topological orders of the nodes and the stages located.

(2) Statistically, in a graph, the output edges of the nodes in the front of topological orders have larger weights. This can be explained that for a node with the order of $i$, the generated $\mathbf{x}_i$ can be received by node $j$ (where $j > i$). This causes the features generated by the front nodes to participate in aggregation as a downstream input. It makes the front nodes contribute more, which can be used to reallocate calculation resources in future work.

(3) From Fig. 6, it can be seen that there exist discrepancies in weight changes for different edges with respect to input samples. The difference is also related to the topological orders of the nodes and the stages located.

(4) Interestingly, in a stage, the edges of the nodes in the back of topological orders have a larger variance. Similarly, the weights of edges in deeper stages also have a larger variance. We speculate that it is related to the level of semantic information of features. Specifically, features generated by the deep layers have high-level semantic information and the correlation of samples is stronger than features with low-level information generated by the shallow layers.

(5) In Fig. 7, it can be seen that for different samples, both the structure and the corresponding weights learned by routers are different. For some "easy" samples, part edges are masked, resulting in a lighter model. This benefits computational efficiency and can be studied further in future work.

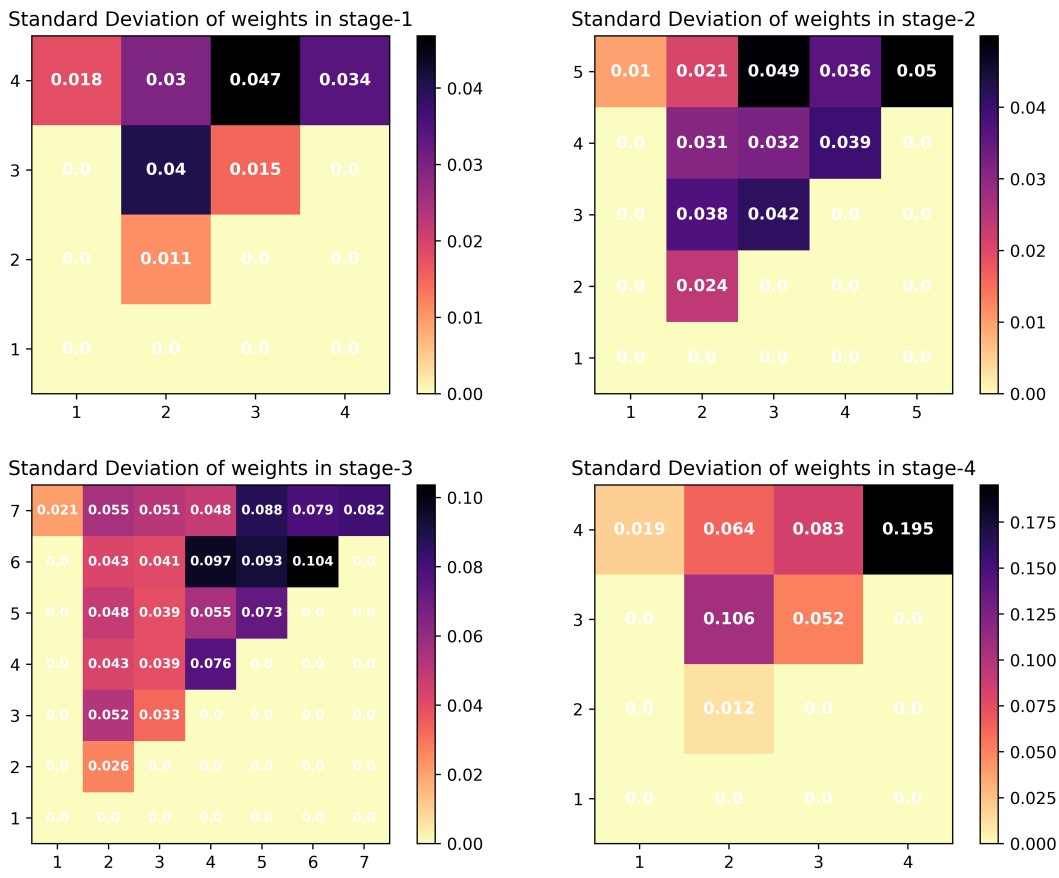

Figure 6: The distribution of standard deviation of weights of edges in different graphs/stages. Darker colors represent larger weights.

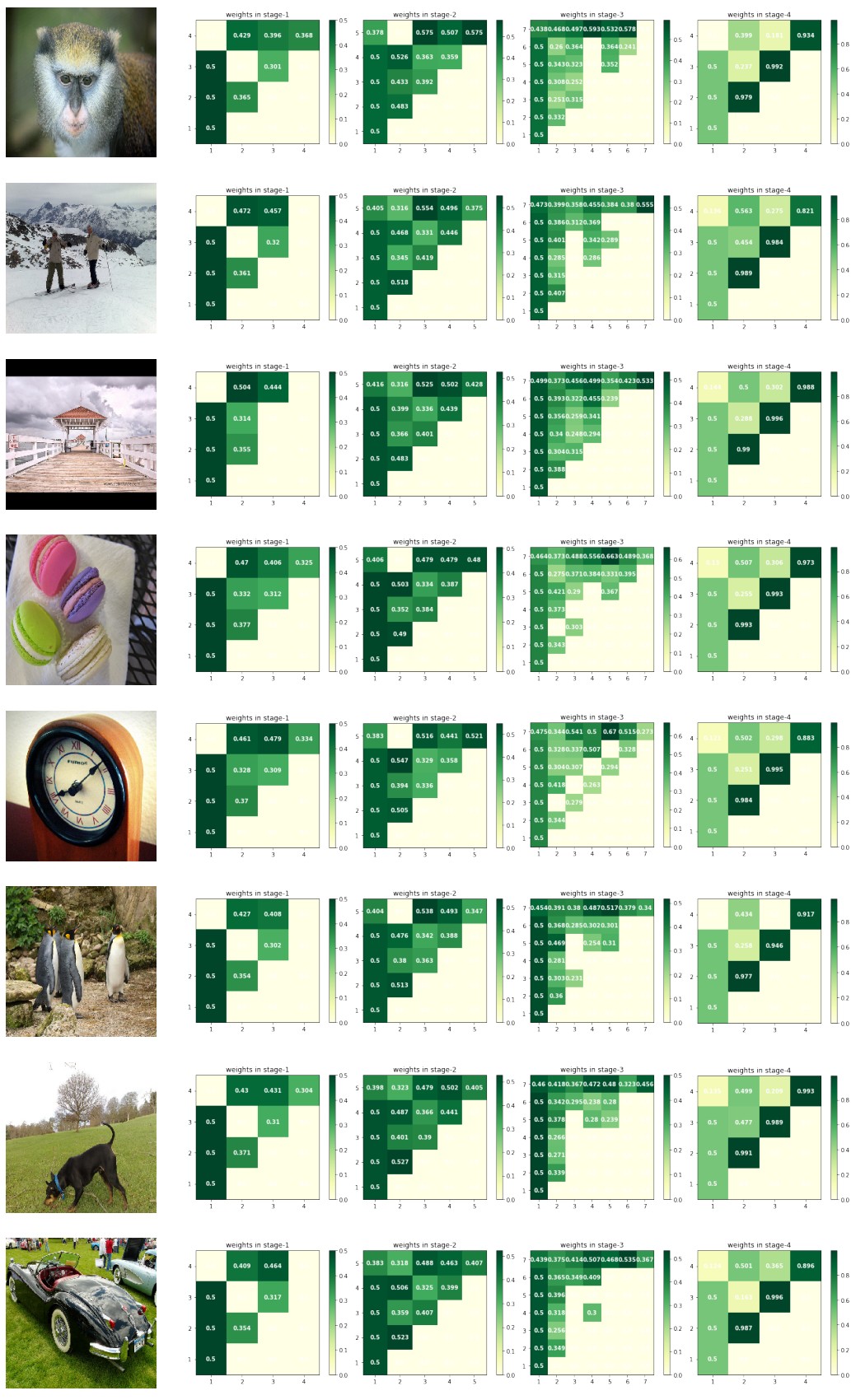

Figure 7: Visualization of the learned connectivity for different input samples.

