# OpenReview forum: "Dynamic Graph: Learning Instance-aware Connectivity for Neural Networks"
_ICLR.cc/2021/Conference — Reject_

### Official Review · AnonReviewer2 · 2020-10-26
**Missing related work**

**Rating:** 6
**Confidence:** 4

**Review:**

This paper presents a novel approach (DG-Net) to “generate” a dynamic structure for the neural network, by learning to predict and select the edges between computational nodes in an end-to-end manner. The method is based on a gating mechanism, applied on top of a fully connected graph (similar to the connectivity in a DenseNet), designed to control the quantity of information received from each previous layer. The experiments show consistent improvement in image classification (ImageNet) and object detection (COCO).

Pro:
- I enjoy the interpretation of the “weighted edges” as a dynamic architecture, able not only to address a more general class of models but also to adapt to each input accordingly (using a second-order approach).
- The method is simple, yet effective with the results on both tasks showing improvement at a low computational cost.
- The paper is clearly written and easy to follow and understand.

Cons:
- In my opinion, two approaches are very related to this paper: the Highway network [1] and the ablations performed in the ResNet paper [2]. In both cases, the intuition is a little bit different: improve the expressiveness by replacing the residual connection with something more powerful - an input-dependent gate. While I agree that this paper offers a more general framework, working on a densely connected graph instead of a limited subset of residual edges, (plus some other minor technical differences: map gates vs scalar gates), the connection to those papers should be clearly discussed in the related work. Also, it is interesting to notice that in [2] this kind of gating decreases the performance compared to the simpler summation, contrary to what we see in DG-Net.
- This idea suggests an additional ablation study: constant-gating instead of dynamic ones (by setting the alpha scalar equals to 1). This is related to the static experiment in the Supplementary material, but more closely to the DenseNet architecture and it would clarify that the improvements come indeed from the learned structure and not from the ability to combine features from different levels.
- It is not clear from the paper how the ResNet architecture is adapted to the dynamic setup. I guess the graph is fully connected on each stage (as long as the spatial dimension is preserved), but since the paragraph about the “multi-stage” architectures comes only in section 3.1, that speaks about classical architectures, not the DG-Net, it is not very clear what densely connected means in the experiments.
- Is any particular initialisation scheme used for the routers? Could it be important for the optimization to start with some parameters that lead to the original version of the model (initialize the parameters such that the residual connections start with alpha=1 and non-existent connections are close to 0)?  For example in [3] all the skip connections are initialised such that the extra module is ignored in the first iterations.

Minors and observations:
- Even if the focus of this work is the dynamic of the edges, it would be interesting to observe if the architecture learns to use all the computational nodes and only connect them differently than what we used to do, or if it rather prefers to drop nodes, thus using a lighter model in the end. (maybe a regularization term that encourages that would be useful for computational efficiency).  - A statistical analysis in this direction could reveal interesting ideas.
- The formatting of Table 1 is hard to follow. At least a vertical line between Baselines and DG-Net would improve the readability.

[1] Highway Networks, Srivastava et. al
[2] Identity mappings in deep residual networks, He et. al, ECCV 2016
[3] Non-local Neural Networks, Wang et. al, CVPR 2018

My main concern regards the connection to previous works that use gating to aggregate information from previous layers, which I see very related to the current work. However, the proposed method is more general and seeing these gates as a routing mechanism that allows the model to learn its own structure in a differentiable way is interesting and could lead to further, more advanced ways of doing this. So, with a clearer discussion of those previous methods and how the current approach differs from them, I lean towards acceptance.

############# UPDATE #############

I thank the authors for their response and for updating the manuscript according to our questions. I agree with the other reviewers that the novelty of this paper is quite limited. However, the idea of using a dynamic, learned graph from a general large search space of models is interesting, provides good empirical results on both image classification and object detection and the authors provide ablation for the new components that motivate the paper. I maintain my initial score: 6.

---

> ### Author Response · Authors · 2020-11-20
> **Response to Reviewer2- Part 2/2**
>
> > Q5: Statistical and visualization analyses of the learned connectivity.
>
> ----
>
> R5: Details are added in Appendix 6.5. To analyze the architecture representations, we visualize the learned connectivity through the adjacency matrices as noted in section 3.4. For the statistical analysis, we show the distribution of the mean of the weights of edges in Fig. 5 and the distribution of standard deviation in Fig. 6. To visually analyze the connections corresponding to different samples, we give the learned connectivity in Fig. 7. Some observations and analyses can be made:
> * From the statistical analysis in Fig. 5, the weights of connections between different nodes have obvious differences. The difference is related to the topological orders of the nodes and the stages located.
> * Statistically, in a graph, the output edges of the nodes in the front of topological orders have larger weights. This can be explained that for a node with the order of $i$, the generated $\mathbf{x}_i$ can be received by node $j$ (where $j > i$). This causes the features generated by the front nodes to participate in aggregation as a downstream input. It makes the front nodes contribute more, which can be used to reallocate calculation resources in future work.
> * From Fig. 6, it can be seen that there exist discrepancies in weight changes for different edges with respect to input samples. The difference is also related to the topological orders of the nodes and the stages located.
> * Interestingly, in a stage, the edges of the nodes in the back of topological orders have a larger variance. Similarly, the weights of edges in deeper stages also have a larger variance. We speculate that it is related to the level of semantic information of features. Specifically, features generated by the deep layers have high-level semantic information and the correlation of samples is stronger than features with low-level information generated by the shallow layers.
> * In Fig. 7, it can be seen that for different samples, both the structure and the corresponding weights learned by routers are different. For some easy samples, part edges are masked, resulting in a lighter model. This benefits computational efficiency and can be studied further in future work.
>
> > Q6: The formatting of Table 1 is hard to follow.
>
> ----
>
> R6: Table 1 has been re-drawn to improve readability.

---

> > ### Comment · AnonReviewer2 · 2020-11-24
> > **RE: Response to Reviewer2**
> >
> > I thank the authors for their response and for updating the manuscript according to our questions. The additional experiments strength the results and clarify my concerns.

---

> ### Author Response · Authors · 2020-11-20
> **Response to Reviewer2- Part 1/2**
>
> We appreciate the suggestions and comments proposed by the reviewers, which lead to further improvements.
>
> > Q1: Comparisons with Highway networks and Identical mapping.
>
> ----
>
> R1: DG-Net is different from Highway networks and Identical mappings in the aspect of search space, modeling type of connections and the learned architectures.
> * Highway networks and ResNet are modified on the plain networks with additional residual connections. The information flow is limited by the number of residual connections. While DG-Net formulates the network into a complete graph, where connections/edges exist between any two layers/nodes. In this way, DG-Net provides more general search spaces that allow more flexible transmission of information flow. Notably, DG-Net overcomes challenges on forwarding computation in section 3.4.
> * In the Highway network, the information flow is determined by a transform gate and a carry gate. These two gates control how much of the output is produced by transforming the input and carrying it. The same definition is also used in the exclusive gating in Identical mapping. However, due to the properties of the complete graph where each node has a direct connection with preceding nodes, DG-Net only requires one transform gate while retaining the original information. Moreover, in the Highway network and the exclusive gating, the sum of the weights for the two gates are constrained to 1. In DG-Net, the weights of different edges are independent of each other. This can preserve the differences in features produced by different nodes.
> * Both Highway networks and identical mapping only change the weights of connections, DG-Net generates instance-aware weights as well as real architectures according to the thresholds. Details are given in Appendix 6.4.
>
> One noteworthy phenomenon is that the conditional gating mechanism decreases the performance in Identical mapping, which is contrary to DG-Net. This can be explained in 2 folds:
> * Under the definition of the exclusive gating, the weight of transformation is defined by $g(x)$, and $1-g(x)$ for identical mapping. But in this case, $g(x)$ approaches 0 will suppress the transformation function. This can be alleviated in the manner of shortcut-only gating. DG-Net overcomes this through the above-mentioned differences.
> * We point out that, according to Eqn. (7) in their paper, that recursively applying the weights of shortcuts will cause information decay and gradient vanishing. This could be the reason why performance drops. In DG-Net, there are direct connections and corresponding weights between layers. And the weights are independent of each other. It avoids recursively applying caused by increasing depth and suppression among gates.
>
> > Q2: Clarification on the source of improvements.
>
> ----
>
> R2: In DG-Net, features from different input nodes are aggregated using a weighted sum. And the channels for different nodes are the same in a stage. Different from DenseNet, this avoids misalignment of channels caused by the removal or addition of edges. If the DenseNet architecture is used, additional modules will be introduced for channel matching, whose computation cost is much larger than that of routers. For a fair comparison, we choose constant gating in Appendix 6.3.
>
> To further analyze the source of the improvement, we do statistical and visualization analyses of the learned connectivity in Appendix 6.5. In Fig. 7, it can be seen that for different samples, both the structure and the corresponding weights learned are different.
>
> > Q3: Dynamic setup for the ResNet.
>
> ----
>
> R3: Yes, the graph is fully connected in each stage to preserve the spatial dimension. This is the same as the static network. We will indicate this in the paper.
>
> > Q4: Different initialization schemes for routers.
>
> ----
>
> R4: In the original paper, the routers are initialized with non-bias. We further conduct different initialization schemes with positive {3} and negative {-3} biases in Appendix 6.4. These initialization methods correspond to existing connections and non-existent connections in the first iterations. Experimental results are given in Fig. 4. It can be seen that the positive initialization of bias achieves lower training loss in the early training procedure and obtains higher validation accuracy of 78.28% (78.00% for the bias of 0, 77.68% for the bias of -3). This suggests that initializing the connections to existing is better than unbiased and non-existent types.
>
> Other related experiments are rerun with bias initialized with 3, resulting in general improvements in Tables 1, 2, 3, 4, 5, 6.

---

### Official Review · AnonReviewer3 · 2020-10-27
**Interesting extension of randomly wired architectures**

**Rating:** 6
**Confidence:** 4

**Review:**

Pros:
- Interesting extension of RandWire to learn better architectures
- Good experimental results

Cons:
- Idea could be seen as minor modification of RandWire
- Doubts about memory requirements

The paper proposes an improve over the idea of randomly wired architectures [1] by exploiting a complete graph where edges are weighted by dynamically computed weights. Dynamically computing edge weights allows the network to optimize its topology. While this idea could be seen as a relatively small modification of [1], it is still very interesting and the results prove its effectiveness in commonly used tasks.

It would have been interesting to show an analysis of what the method is actually learning, e.g. if a particular topology emerges from the adjacency matrix, in order to provide more insights on why the method is effective.

I also have a concern about memory requirements that I would like the authors to address. It is true that DG-Net does not require significantly more FLOPS or trainable parameters. However, you use a complete graph which means that it requires to store in memory O(L^2) activation tensors, i.e. the activation of each convolutional layer weighted by edge weight. This could become a limitation for larger networks. Note that this is also a problem for [1] but, in that case, the sparse connectivity mitigates the issue.


[1] Saining Xie, Alexander Kirillov, Ross B. Girshick, and Kaiming He.  Exploring randomly wired neural networks for image recognition.

---

> ### Author Response · Authors · 2020-11-20
> **Response to Reviewer3 - Part 2/2**
>
> > Q3: Memory requirements in optimization and inference.
>
> ----
>
> R3: We give the formula of gradient update in section 3.5 of Eqn. (7~10). Compared with static networks, the main additional memory during training is required by the aggregated features $\mathbf{x}^{j^\prime}$ for the purpose of back-propagation. It is related to the number of nodes in a graph. And the used \textit{addition} operation of feature aggregation can effectively save memory instead of concatenation. We also test the actual memory increase during optimization. For ResNet50, the static network requires ~6G memory and ~8G for the dynamic one. For ResNet-101, the static network needs ~9G memory and ~12G for the dynamic one. DG-Net roughly requires an additional 30% memory during optimization. For much larger networks, some techniques (e.g. checkpoints and model parallel) can be used for training.
>
> Since the additional memory requirements are caused by the gradient update, this additional memory is not needed during inference. For ResNet50, the memory requirements are 3242M(static) and 3512M(dynamic) during inference. For ResNet101, the memory requirements are 3280M(static) and 3621M(dynamic) during inference. The additional memory requirements are acceptable compared to the improvements, which allows the deployment of resource-limit devices.

---

> ### Author Response · Authors · 2020-11-20
> **Response to Reviewer3 - Part 1/2**
>
> We thank the reviewers for their suggestions and comments, which can help obtain some additional insights.
>
> > Q1: Differences with RandWire.
>
> ----
>
> R1: DG-Net is largely different from RandWire in the aspect of search method, definition of search space, modeling method, performance, and conclusions. It can be summarized in 5 folds:
>
> * RandWire aimed to find graph generators that can produce new families of random models for further selection. And the hyper-parameters for the graph generators and generated architectures are selected by trial-and-error by human designers. While DG-Net intends to optimize the connectivity in a differentiable way, which is more effective and task-related.
> * RandWire defines the search spaces according to the properties of the graph generator. Once the graph is generated, only the weights of edges are learned. While DG-Net initializes the search space as complete graphs, which contains all possible connections. During training, both the architecture and the weights of connections can be learned, resulting in much larger search spaces.
> * RandWire assigns each edge with a learnable weight. During inference, the weight is used for all samples. In DG-Net, the weights are generated based upon inputs, allowing more flexible connections. The proposed instance-aware connectivity also overcomes challenges on modeling and forward computation. A router along with the node is used to generate weights for the output edges, which is effective and efficient (discussed in Appendix 6.2). DG-Net requires the aggregation of features differently for each input sample. The proposed buffer mechanism (in section 3.4) successfully solves this without introducing excessive computation or time-consuming burden.
> * We compare the performance in Table 4. The mean and standard deviation of results are given through 5 repeat runs. DG-Net obtains higher performance (73.52 vs 71.34/71.16/72.26) under the small computation regime proposed in section 4.2 of their paper. More importantly, the result of DG-Net is more stable with a small deviation (0.05 vs 0.40/0.34/0.27). RandWire largely suffers from randomness even under the best setting in their paper. This phenomenon is consistent with Fig. 3 in their paper, where large variances exist under the same setting of graph generators.
> * RandWire claimed that random graphs generated by well-defined graph generators are good enough. While DG-Net demonstrates that the connectivity for each sample can be optimized, resulting in better performance than the static architectures.
>
> | wiring type| top-1 |
> | ---- | ---- |
> | ER(P=0.2) | $71.34\pm 0.40$ |
> | BA(M=5) | $71.16\pm 0.34$ |
> | WS(K=4, P=0.75) | $72.26\pm 0.27$ |
> | DG-Net |$73.52\pm 0.05$ |
>
> > Q2: Statistical and visualization analyses of the learned connectivity.
>
> ----
>
> R2: Details are added in Appendix 6.5. To analyze the architecture representations, we visualize the learned connectivity through the adjacency matrices as noted in section 3.4. For the statistical analysis, we show the distribution of the mean of the weights of edges in Fig. 5 and the distribution of standard deviation in Fig. 6. To visually analyze the connections corresponding to different samples, we give the learned connectivity in Fig. 7. Some observations and analyses can be made:
> * From the statistical analysis in Fig. 5, the weights of connections between different nodes have obvious differences. The difference is related to the topological orders of the nodes and the stages located.
> * Statistically, in a graph, the output edges of the nodes in the front of topological orders have larger weights. This can be explained that for a node with the order of $i$, the generated $\mathbf{x}_i$ can be received by node $j$ (where $j > i$). This causes the features generated by the front nodes to participate in aggregation as a downstream input. It makes the front nodes contribute more, which can be used to reallocate calculation resources in future work.
> * From Fig. 6, it can be seen that there exist discrepancies in weight changes for different edges with respect to input samples. The difference is also related to the topological orders of the nodes and the stages located.
> * Interestingly, in a stage, the edges of the nodes in the back of topological orders have a larger variance. Similarly, the weights of edges in deeper stages also have a larger variance. We speculate that it is related to the level of semantic information of features. Specifically, features generated by the deep layers have high-level semantic information and the correlation of samples is stronger than features with low-level information generated by the shallow layers.
> * In Fig. 7, it can be seen that for different samples, both the structure and the corresponding weights learned by routers are different. For some easy samples, part edges are masked, resulting in a lighter model. This benefits computational efficiency and can be studied further in future work.

---

### Official Review · AnonReviewer4 · 2020-10-28
**Nice idea, but limited novelty and experimental validation**

**Rating:** 6
**Confidence:** 3

**Review:**

This work proposes a novel method, called Dynamic Graph Network (DG-Net), for optimizing the architecture of a neural network. Building on the previous work introduced by (Xie et al., 2019), the authors propose to consider the network as a complete directed acyclic graph (DAG). Then, the edge weights of the DAG are generated dynamically for each input of the network. At each node of the network, the authors introduce an extra-module, called router, to estimate the edge weights as function of the input features.

The proposed method addresses the problem of optimizing the connectivity of neural networks in an interesting way, where the architecture is not fixed but it depends on the input instances. Moreover, I think that a strong advantage of the proposed technique is that the optimization of the architecture comes with a negligible extra cost both in terms of parameters and computational complexity. Overall, the paper is well written and easy to follow.

My only serious concern is the degree of novelty with respect to (Yuan et al., 2020), which was published at ECCV 2020. The main difference seems to be that in the proposed method the graph is dynamic (i.e., it depends on the input instances), instead  in (Yuan et al., 2018) the graph is learned but fixed for all the input samples. In the experimental results, I would have expected a deeper ablation study on the importance of the dynamic graph, since this is the main contribution of the paper. Instead, there is only one experiment in the appendix (Table 6). Therefore, the impact of the dynamic graph in the performance of the proposed method is not clear and it is difficult to evaluate the importance of this contribution.

Other comments:

- In Sec. 3.1 the authors say that the ResNet architecture can be represented with a DAG where the set of edges is defined as E={(i,j)|j=i+1,i+2}. This is not true: if you unroll the definition of the ResNet architecture, as done in Eq. (4)-(6) in [1], and compare it with what you obtain using Eq. (1), it is easy to see that the two resulting functions are different.

- The definition of the convolutional block is not clear, is it a ReLU-conv-BN triplet as in (Xie et al., 2019)?

- The use of a DAG with edge weights for representing the architecture is not novel, it was already introduced in (Xie et al., 2019).

- In Sec. 4.3, Table 5 shows a comparison with state-of-the-art NAS-based methods. DG-Net is implemented using RegNet-X and RegNet-Y as the basic architecture, however in Table 5 the performance of the basic architectures (without the dynamic graph optimization) is not reported, this would be useful to evaluate the gain provided by the optimization of the architecture.


[1] Veit et al., Residual Networks Behave like Ensembles of Relatively Shallow Networks, NIPS 2016

###############################################################################################

After the discussion period:

I thank the authors for their responses and for updating the paper. The authors have added a deeper analysis on the impact of the dynamic graph, however I still believe that the novelty of the paper is a bit limited. I have slightly increased my score to 6.

---

> ### Author Response · Authors · 2020-11-20
> **Response to Reviewer4**
>
> We thank the reviewers for their suggestions and comments, which can help improve the paper.
>
> > Q1: About the novelty and comparisons with (Yuan et al., 2020).
>
> ----
>
> R1: Compared with (Yuan et al., 2020), DG-Net is different in 5 folds:
>
> * (Yuan et al., 2020) tried to learn a fixed topology for the network, whose method is searching for the weights of edges in a complete graph. During inference, the weights for different samples are the same. In DG-Net, the weights of edges are generated based upon inputs, allowing more flexible connectivity patterns.
> * Due to the above advantage, DG-Net owns a larger search space than (Yuan et al., 2020). Their method is limited by the size of search space and cannot achieve significant improvement in small spaces (in their paper section 4.1, 4.2). This limits the application on some classic networks where each stage only has 3~6 layers. DG-Net overcomes this shortcoming and has been verified on many architectures. DG-Net surpasses baselines in large margins up to 1.61% in ImageNet in Table 1.
> * The proposed instance-aware connectivity also overcomes challenges on modeling and forward computation. A router along with the node is used to generate weights for the output edges, which is effective and efficient (discussed in Appendix 6.2). DG-Net requires the aggregation of features differently for each input sample. The proposed buffer mechanism (in section 3.4) successfully solves this without introducing excessive computation or time-consuming burden.
> * Instead of only learning the weights of edges in (Yuan et al., 2020), DG-Net also adjusts the real architectures according to the learned thresholds, which can bring better acceleration. We also add the statistical and visualization analyses of the learned connectivity in Appendix 6.5, which bring some interesting observations and may lead to future work.
> * Last, DG-Net achieves obviously better experimental results than (Yuan et al., 2020). We update the ablation study in Appendix 6.3, which reflects the improvements brought by the dynamic graph itself. In classical networks of ResNet-18/50/101 and MobileNet-v2, the proposed dynamic graph outperforms static networks in large margins. Compared with networks with learnable weights of complete graphs, DG-Net also demonstrates the effectiveness of the instance-aware connectivity.
> And the results also prove that DG-Net overcomes the limitation of search spaces met in (Yuan et al., 2020).
>
> | backbone | method | top-1 | gain |
> | ---- | ---- | ---- | ---- |
> | ResNet-18 | baseline | 70.30 | - |
> | ResNet-18 | leanable weights | 70.51 | +0.21 |
> | ResNet-18 | dynamic graph | 71.32 | +1.02 |
> | ResNet-50 | baseline | 76.70 | - |
> | ResNet-50 | leanable weights | 77.00 | +0.30 |
> | ResNet-50 | dynamic graph | 78.28 | +1.58 |
> | ResNet-101 | baseline | 78.29 | - |
> | ResNet-101 | leanable weights | 78.64 | +0.35 |
> | ResNet-101 | dynamic graph | 79.90 | +1.61 |
> | MBNet-v2-1.0| baseline | 72.60 | - |
> | MBNet-v2-1.0| leanable weights | 72.86 | +0.26 |
> | MBNet-v2-1.0| dynamic graph | 73.54 | +0.94 |
>
> > Q2: DAG representation of ResNet.
>
> ----
>
> R2: Thanks to the reviewer for pointing this out. We show the nature view of ResNet as E={(i,j)|j=i+1,i+2}. Under the unrolled type in (Veit et al., 2016), the representation can be denoted as the wiring pattern as DenseNet. We have corrected this in the paper.
>
> > Q3: The definition of the convolutional block is not clear, is it a ReLU-conv-BN triplet as in (Xie et al., 2019)?
>
> ----
>
> R3: Yes. We follow the definition in (Xie et al., 2019) which allows the aggregation to receive both positive and negative activation, preventing the aggregated activation from being inflated in case of a large input degree. We will add this to the paper.
>
> > Q4: The use of a DAG with edge weights for representing the architecture is not novel, it was already introduced in (Xie et al., 2019).
>
> ----
>
> R4: The difference is the way of modeling the weights of edges. And DG-Net is also largely different from randwire in motivation, the capacity of search space, performance, and conclusions. Details are given in the Response to Reviewer3 of R1.
>
> > Q5: The original performance of RegNet-X and RegNet-Y in Table 5.
>
> ----
>
> R5: We add the original performance under the same training setting. For RegNet-X-600M, the accuracy is 75.03%. DG-Net obtains 0.81% improvement. For RegNet-Y-600M, the accuracy is 76.1%. DG-Net improves accuracy by 0.90%. Since the architectures are already the best in a search space with $10^{18}$ possible configurations, the results are considerable.

---

### Official Review · AnonReviewer1 · 2020-10-28
**A radical new architecture, but unfortunately the actual design is not fully convincing and based on strong theoretical foundations.**

**Rating:** 3
**Confidence:** 4

**Review:**


The paper discusses a model for learning the architecure of a convolutional networks starting from a fully connected graph. The idea is to learn the adjacency graph of the model together with the weights of the networks.

Strenghts:

- The idea of thinking out-of-the-box by imagining new architectures is very attractive and interesting.

Weaknesses:

- The actual advantages in the model do not look apparent given the results in the evaluation section.

- The theoretical foundation of this work is unclear. For example, it is unclear how the proposed solution will work in practice in terms of back-propagation.

- The performance results show that the proposed method is characterized by performance close to those of existing methods.


In general, I really welcome this type of work: non-conventional, experimental and quite radical in terms of approach. However, unfortunately, the authors do not provide a convincing description of their approach. Unfortunately, it does not appear that the method is developed on a sufficiently strong theoretical basis. For example, it is unclear how back-propagation work in these circumstances when you don't have a stacked architecture.

The choice of the thresholds and the actual learning of the adjacency matrix is not described in sufficient detail.

The actual computation complexity and the trade-offs in terms of computational complexity/accuracy is unclear.

In the experimental results, the actual performance of the method appear very similar to the other methods. In some cases they might be the same since the confidence intervals are overlapping.

Questions:

- What is the theoretical foundation of the method? Given the fully connected nature of the graph, how does back-propagation work in this case? In a sense, in fact, you have a DAG, how do you deal with cycles? When you do you stop the back-propagation if you do not have a stacked architecture?

- Could you please provide the confidence intervals for all the results you presented? In fact, it seems that the values related to your approach are better than existing techniques in some cases but they look very close?

- Is the computational complexity justified? Also note: you probably need a larger number of samples to learn the additional adjacency graph. What is the trade-off? Given the gain in terms of performance, the actual additional complexity might not be completely justified.

---

> ### Author Response · Authors · 2020-11-20
> **Response to Reviewer1**
>
> We thank the reviewers for their suggestions and comments.
>
> > Q1: What is the theoretical foundation of the method? Given the fully connected nature of the graph, how does back-propagation work in this case?
>
> -------
>
> R1: We have added the update rules in terms of back-propagation for DG-Net in section 3.5 of Eqn. (7~10). The parameters of the convolutional operations and routers can be optimized jointly in a differentiable way.
> Notably, the threshold for each router is also learnable through acting as $\alpha \cdot \sigma(\alpha - \tau)$, whose gradient can be computed by $\sum (\frac{\partial {L_t}}{\partial {x^j}} \odot \frac{\partial {f^j}}{\partial {x^{j^\prime}}} \odot \mathbf{x}^i) \cdot \frac{\partial {\psi}^j}{\partial \tau^j}$. It avoids careful selections of hyper-parameters for the thresholds in practice. We also visualize the actual learned connectivity for different input samples in Fig. 7. In the adjacency matrix, each element is generated by the corresponding router based upon inputs. The gradients w.r.t each element $\alpha^{(i,j)}$ can be noted as $\sum (\frac{\partial \mathcal{L}_{t}}{\partial \mathbf{x}^j} \odot \frac{\partial {f}^j}{\partial \mathbf{x}^{j^{\prime}}} \odot \mathbf{x}^i)$.
>
> > Q2: In a sense, in fact, you have a DAG, how do you deal with cycles? When do you stop the back-propagation if you do not have a stacked architecture?
>
> -------
>
> R2: DG-Net discusses the network structure belonging to DAGs (Directed Acyclic Graph),  which do not have cycles. The graphs that include cycles are not included in this paper.
>
> > Q3: Could you please provide the confidence intervals for all the results you presented? In fact, it seems that the values related to your approach are better than existing techniques in some cases but they look very close?
>
> -------
>
> R3: DG-Nets achieve significant improvements compared with existing methods, especially in the large-scale datasets of ImageNet and COCO. Added repeat runs also prove the stability of DG-Net. The comparisons can be analyzed in 4 folds:
>
> * In Table 1, DG-Net surpasses baselines in large margins up to 1.61% in ImageNet under **the same training setting**. In Table 2, DG-Net also obtains up to 2.73% gains in COCO object detection. Due to the training cost in ImageNet, we report part confidence intervals of networks, including MobileNetv2-1.0 ($73.54\pm 0.06$) resnet18 ($71.32\pm 0.14$), resnet50 ($78.28\pm 0.04$) through 5 repeat runs. Results show consistent improvements with a small variance. We will update Table 1 by adding confidence intervals.
>
> * In Table 3, DG-Net also outperforms InstaNAS (73.2% vs 71.9%) under similar latency constraints with small variance (0.06) through 5 repeat runs.
>
> * In Table 4, we conduct 5 repeat runs for graph generators of ER($71.34\pm 0.40$), BA($71.16\pm 0.34$), WS($72.26\pm 0.27$) and DG-Net($73.52\pm 0.05$). It can be seen that randwire suffers from randomness caused by their random graph generators. The same phenomenon can be observed from their paper in Fig. 3. DG-Net can achieve more stable results with small variance (0.05).
>
> * In Table 5, we add the comparison in search cost of different NAS methods. Since the architecture in DG-Net can be optimized in a differentiable manner, higher results can be obtained with less search cost.
>
> > Q4: Is the computational complexity justified? Also note: you probably need a larger number of samples to learn the additional adjacency graph. What is the trade-off? Given the gain in terms of performance, the actual additional complexity might not be completely justified.
>
> -------
>
> R4: The training configurations of all comparison experiments are the same as the baselines, which means the dynamic graph is optimized using the same number of samples with static networks. Moreover, DG-Net does not require significantly more FLOPs or trainable parameters as shown in Tables 1,2.
>
> We want to emphasize that the performance improvements come from the appropriate connection method for each sample, not the computational complexity. And the results in Table 6 can prove this. For the network with $\alpha$, all connections exist with larger computational cost. But DG-Nets can also outperform them in less computational cost with only critical connections.

---

### Author Response · Authors · 2020-11-21
**Paper Update**

Thanks to all the reviewers for their constructive suggestions and comments.

In this updated version, we carefully considered all reviewers’ suggestions to improve the paper. Generally, we performed 6 aspects of works:
* The gradient update rules are given in section 3.5 for the parameters of convolutional operations and routers.
* Instead of applying the non-bias initialization scheme for the biases of routers, we initialize the biases with positive ones of 3, resulting in the existence of connections in the first iterations. Through an ablation study on different initialization schemes in Appendix 6.4, we find the positive one is better. So we retrain related experiments in ImageNet and COCO, results are generally improved and updated in Tables 1,2,3,4,5.
* More detailed comparisons with different connectivity methods are given in Appendix 6.3, which further verify the improvements brought by dynamic graphs themselves.
* Repeat runs are conducted in Tables 3,4. Results show consistent improvements with a small variance compared with related methods..
* Detailed statistical and visualization analyses are given in Appendix 6.5, which demonstrate the distributions of the learned connectivity. And some interesting phenomena have been observed, which can inspire future work.
* The formatting of tables has been re-drawn to improve readability.

---

### Decision · Program_Chairs · 2021-01-07
**Final Decision**

**Decision:**

Reject

**Comment:**

The idea presented in the paper is interesting and has caught the attention of the reviewers. However there seem to be only a tepid support for acceptance with a reviewer championing rejection.
There is little novelty in the approach but empirical validation shows results that consistently improve over selected baselines. I am afraid that more evaluations would be needed at this stage to consider this work for acceptance.